# The relationship between precipitation and insurance data for floods in a Mediterranean region (Northeast Spain)

5  Maria Cortès[1,2], Marco Turco[1], Montserrat Llasat-Botija[1,2], M. Carmen Llasat[1,2]

[1]Department of Applied Physics, University of Barcelona, Barcelona, 08028, Spain
[2]Water Research Institute (IdRA), University of Barcelona, Barcelona, 08001, Spain

*Corresponding author*: Maria Cortès, Department of Applied Physics, University of Barcelona, Martí I Franqués 1, 08028 Barcelona, Spain, mcortes@meteo.ub.edu

# The relationship between precipitation and insurance data for floods in a Mediterranean region (Northeast Spain)

Maria Cortès[1,2], Marco Turco[1], Montserrat Llasat-Botija[1,2], M. Carmen Llasat[1,2]

[1]Department of Applied Physics, University of Barcelona, Barcelona, 08028, Spain
[2]Water Research Institute (IdRA), University of Barcelona, Barcelona, 08001, Spain

*Correspondence to*: Maria Cortès (mcortes@meteo.ub.edu)

**Abstract.** Floods in the Mediterranean region are often surface water floods, in which intense precipitation is usually the main driver behind the events. Determining the link between the causes and impacts of floods can make it easier to calculate the level of flood risk. However, up until now, the limitations in quantitative observations for flood-related damages have been a major obstacle when attempting to analyse flood risk in the Mediterranean. Flood-related insurance damage claims for the last 20 years could provide a proxy for flood impact, and this information is now available in the Mediterranean region of Catalonia, in northeast Spain. This means a comprehensive analysis of the links between flood drivers and impacts is now possible. The objective of this paper is to develop and evaluate a methodology to estimate flood damages from heavy precipitation in a Mediterranean region. Results show that our model is able to simulate the probability of a damaging event as a function of precipitation. The relationship between precipitation and damage provides insights into flood risk in the Mediterranean and is also promising for supporting flood management strategies.

## 1 Introduction

Flooding is the main natural risk in the world. Between 2005 and 2014, more than 85,000,000 people were directly affected by flood events annually, and around 6,000 people were killed on average each year due to floods (UNISDR, 2015). The main factors involved in flood risk analysis are the hazard, or the likelihood of a natural phenomenon causing damages, and the vulnerability, that is, the characteristics and circumstances of a community/system that make it susceptible to potential flood damage (UNISDR, 2009; Kundzewicz et al., 2014; Winsemius et al., 2015). Vulnerability can include factors such as exposure and other societal factors such as early warning systems, building capacity to cope with natural hazards, and disaster recovery infrastructure (Jongman et al., 2014; Nakamura and Llasat, 2017).

A large number of authors are making efforts to create methodologies that are able to analyse the impacts of floods, due to the significant consequences of this phenomenon (Messner and Meyer, 2006; García et al., 2014). Indeed, progress is being made on incorporating impact and vulnerability analysis in flood risk assessment, although the limitations of the impact data (availability and quality) make it difficult to carry out these studies (Elmer et al., 2010; Petrucci and Llasat, 2013; Jongman et al., 2014; Papagiannaki et al., 2015; Thieken et al., 2016; Kreibich et al., 2017).

Insurance data may provide a good proxy for describing flood damages (Barredo et al., 2012). Several recent works have used this kind of data to explore the causes and impacts of floods. For instance, in several European regions researchers have noted that precipitation has a significant influence on flood insurance data (see for instance Spekkers et al., 2013, 2015 for Netherlands; Zhou et al., 2013 for Denmark; Sampson et al., 2014 for Ireland; Moncoulon et al., 2014 for France; Torgersen et al., 2015 for Norway). This data is very valuable for establishing causal relationships between the costs of flood damage and precipitation extremes, for developing risk maps, and to use as a validation tool for damage models (Zhou et al., 2013). These studies agree on the potential of insurance data to assess the damage caused by pluvial and urban floods.

Most floods that have affected the region of study, Northeast Spain, are surface water floods that caused catastrophic damage (Llasat et al., 2014, 2016a). This type of floods can be regarded as coming under the most general definition of rainfall-related floods (Bernet et al., 2017), including pluvial floods but also flooding from sewer systems, small open channels, diverted watercourses or flooding from groundwater springs (Falconer et al., 2009). River floods that affect great distances are very rare in the region, and are only related to catastrophic and extended floods (for the analysed period only the October 2000 floods were of this type). Nevertheless, these are usually absorbed by reservoirs. It is therefore expected that flood insurance data will correlate strongly with precipitation and surface water floods. However, relatively few studies exist for the Mediterranean region, being mostly limited to urban flood damage assessment (Freni et al., 2010; Papagiannaki et al., 2015; Bihan et al., 2017), while an analysis of the possible links between precipitation and economic flood damages are yet to be assessed across Mediterranean regions. This may be due to limitations in insurance data records and difficulties in estimating how heavy precipitation could affect monetary damages. In the Mediterranean region of Catalonia, in Northeast Spain, 20 years of flood-related insurance damage claims are available from the Spanish public reinsurer, the "Insurance Compensation Consortium" (*Consorcio de Compensación de Seguros*, or CCS), a public institution that compensates homeowners for damage caused by floods, which plays a role similar to that of a reinsurance company (Barredo et al., 2012). This means an assessment of the links between precipitation and impacts is now possible. This analysis would greatly help policy-makers and civil protection agencies, improving early warning systems and allowing for more efficient management strategies. Furthermore, assessing the relationship between precipitation and flood damages would provide relevant information on the mechanisms behind how floods evolve, as well as the underlying mechanisms in Mediterranean regions.

The aim of this study is to develop and evaluate a methodology to estimate surface water flood damages from heavy precipitation in the Mediterranean region of study (from now on, we will use the expression "flood" to refer to surface water floods). The relationship between precipitation and insurance data has been assessed, using logistic regression models, to assess the probability of large monetary damages in relation to heavy precipitation events. Specifically, our main goal is to answer the following research questions:

1. Can we predict flood damages with parsimonious precipitation-damage models?
2. To what extent do exposure and vulnerability (through the commonly used proxies of Gross Domestic Product), and population (Pielke and Downton, 2000; Choi and Fisher, 2003; Barredo, 2009) determine the damages corresponding to precipitation events?

3. Which thresholds used to define large flooding damages and heavy rainfall events determine the best applicability range?

To sum up, the results of this study can help better understand flood risk in Mediterranean areas by analysing flood causes and impacts, and can help more accurately estimate flood damage when high rainfall is forecast.

The study is organised as follows. After the Introduction, the section on "Methods" describes the study region, the observed data and the methodology used. Then, the "Results" section presents the regression models obtained. Finally, the "Conclusions" section summarises the main findings of this study.

## 2 Methods

### 2.1 Study region

The study area is Catalonia, a Spanish region of 32,108 km$^2$ in the northeast Iberian Peninsula. The region is characterised by three mountain ranges (Fig. 1): the Pyrenees in the north (maximum altitude above 3,000 MASL) and parallel to the Mediterranean coast (SE-NE) between the Pre-Littoral mountain range (maximum altitude around 1,800 MASL) and the Littoral mountain range (maximum altitude around 600 MASL). This marked orography is the key reason for the development of floods, both from a hydrological point of view (small torrential catchments) and due to meteorological factors (the orography

forces water vapour to rise from the Mediterranean, triggering instability; Llasat et al., 2016a). The region is divided into 42 counties and 948 municipalities, with a total population of 7.5 million, most of them living along the coast, where more than 70% of the flood events occur (Llasat et al., 2014), making it a very vulnerable area. From a hydrological point of view the region is divided into 31 basins, most of them with surface areas of less than 500 km$^2$. Some of them are formed by very small municipalities for which some data needed is not available (i.e. Gross Domestic Product, GDP). For this reason we have

aggregated some of the basins and worked with a total of 29 (see supplementary material).

We also analyse the Metropolitan Area of Barcelona (MAB, 534.7 km$^2$) (Fig. 1) in detail, which consists of the city of Barcelona (1,608,746 inhabitants in 101.3 km$^2$) and 35 municipalities. Although it represents less than 2 % of the surface area of Catalonia, it contains 48 % of the population (IDESCAT, 2016). It is affected by an average of over 3 flood events per year, most of which are floods due to very convective local precipitation (Cortès et al., 2017). The city of Barcelona is crossed by

20 streams that have their source in the Serra de Collserola (Littoral mountain range), and which are covered as part of the Barcelona drainage system, managed by the Barcelona Water Cycle (*Barcelona Cicle de l'Aigua* or BCASA). The United Nations International Strategy for Disaster Reduction (UNISDR) marked Barcelona as a resilient city and a model city for dealing with floods (Nakamura and Llasat, 2017), as it has a permanent surveillance and warning system running on hydraulic modelling that includes 15 rainwater tanks (13 underground and 2 open) that allow for better flood prevention. As a result,

flood damages have decreased over time (Barrera-Escoda et al., 2006) while the daily rainfall threshold associated with damaging floods has increased (Barrera-Escoda and Llasat, 2015).

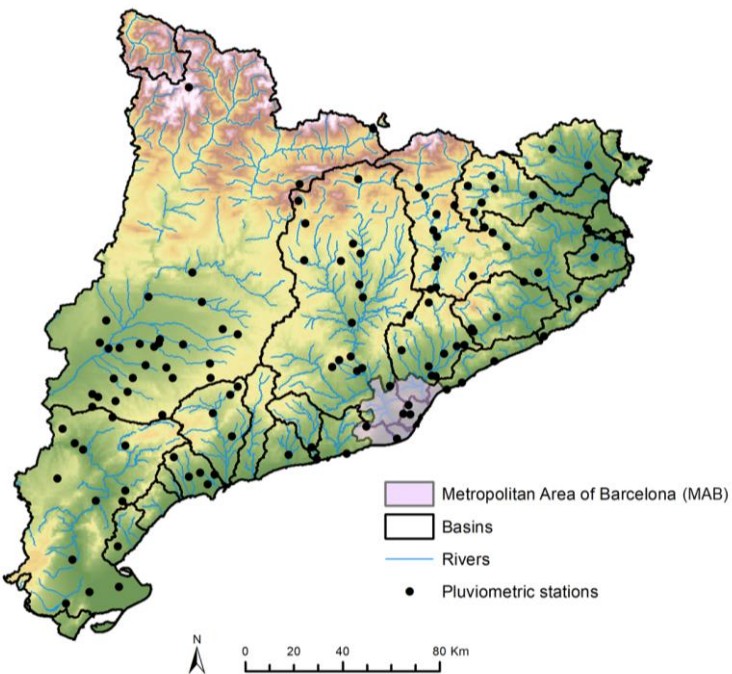

**Figure 1: Map of Catalonia showing the aggregated basins, the Metropolitan Area of Barcelona (MAB), the main rivers and the pluviometric stations used.**

## 2.2 Data

The flood damage data were obtained from the insurance compensations for floods paid by the Spanish Insurance Compensation Consortium (CCS). The CCS compensates for damages caused to people and property by floods and other adverse weather events covered by an insurance policy. The CCS database includes more than 58,000 records of claims paid for floods in Catalonia, provided on a postal code level for the 1996-2015 period (no previous information is available with this level of detail). For flood events we use the INUNGAMA (Barnolas and Llasat, 2007; Llasat et al., 2016a) and PRESSGAMA (Llasat et al., 2009) databases, which report the flood events that have occurred in Catalonia on a municipal, district and basin level. Basic data on damaging events (i.e. event dates, duration and some hydrometeorological data) are identified using the INUNGAMA database. The PRESSGAMA database was used for the events and the description of their impacts, and to identify the worst-affected places. Population and Gross Domestic Product data were obtained from the Statistical Institute of Catalonia (*Institut d'Estadística de Catalunya*, IDESCAT). The population and GDP used correspond to the year when the flood event took place. We use daily precipitation data provided by the meteorological station network run by the Spanish State Meteorological Agency (*Agencia Estatal de Meteorología,* or AEMET). To ensure temporal homogeneity, we have only considered the stations located in Catalonia with more than 90 % of valid data over the 1996-2015

period (Fig.1). For the MAB we also considered 30-minute weather data obtained from the network of automatic meteorological stations belonging to the Meteorological Service of Catalonia (*Servei Meteorològic de Catalunya,* or SMC). Table 1 summarises the data used.

Compensations paid by the CCS were adjusted to the value of the euro in 2015, following the methodology defined by the Spanish National Institute of Statistics (INE, 2007). This consists of using the exchange rate in the Consumer Price Index (CPI) between the two years to adjust the values shown in euros. To compare this data with other variables, we first aggregated them at a municipal level. This task was made more difficult by the fact that a municipality can include different postcodes and one postcode can correspond to two municipalities. These difficulties were solved by aggregating the municipal postcodes and looking at press information. Finally, to calculate the total damages per event, we took the payments made on the day the event occurred, and the following seven days. We used this seven-day window as this is the period of time that the CCS allows insurance claims to be made. When the time difference between two events is less than seven days, damages are associated with the first event, if the date of the claim was before the first day of the second event.

Because the available data are too sparse to support our statistical assessment on a municipal scale, we assessed the precipitation-compensation link for Catalonia as a whole. That is, we sampled pairs of the response variable (i.e. the compensation series) and the maximum 24 h precipitation for each basin recorded, and pooled them into one sample for the entire region (Catalonia) to correlate them. For each event there can be more than one pair of values, depending on the number of affected catchments. From now on we will use the expression "flood case" for each pair of values corresponding to a basin affected by a flood event. This area is large enough to have a fairly large sample size for analysis, but small enough that the causes of flood damages are likely to be similar across the area.

The same methodology was applied for another spatial aggregation based on the Spanish State Meteorological Agency (AEMET) warning areas (included in the supplementary material), and which has also been used in other studies like Quintana-Seguí et al. (2016). Similarly as for the basins, an aggregation process was carried out (15 to 14 warning areas).

Finally, we considered three categories of damages: (i) total damages (D), (ii) damage per capita (DPC) and (iii) damage per unit of gross domestic product (DPW). This meant the relative impacts of socio-economic factors on damage could be estimated, while taking into account population and wealth (Zhou et al., 2017).

## 2.3 Modelling damage probabilities

After gathering together a list of all the floods that affected Catalonia between 1996 and 2015, we filtered them based on specific rainfall thresholds. The Social Impact Research Group, created within the framework of the MEDEX project (MEDiterranean EXperiment on cyclones that produce high-impact weather in the Mediterranean; http: //medex.aemet.uib.es) has established a threshold – when a maximum rainfall of over 60 mm in 24 h was recorded – to indicate the expected social impact for rain events in Catalonia (Amaro et al., 2010; Jansà et al., 2014). Barbería et al. (2014) suggest that the threshold of 40 mm/24 h is better for urban areas. In the main text we consider the threshold on 60 mm/24 h, while results obtained using the lower threshold are available in the supplementary material.

In the case of the MAB, the minimal unit of study is the entire MAB region, which means each flood event corresponds to a single flood case. Taking into account that applying the precipitation thresholds of 40 and 60 mm for the MAB will result in samples that are too small (36 and 23 flood cases, respectively), and that the analysis would not be robust enough, we have used lower precipitation thresholds. It is worth noting that in this case we also used 30 min precipitation, which means a lower

threshold might still have significant consequences. For instance, a previous study shows that with precipitation over 20 mm/30 min, extraordinary and catastrophic flood events can occur (Cortès et al., 2017) in the region. In addition, other studies (Barrera-Escoda and Llasat, 2015) have used 20 mm/24 h to study flood events in this Mediterranean region. Since the sample size is still small, a 10 mm threshold was also used (but results for the 20 mm threshold are available as supplementary material).

Figure 2 shows the relationship between the three categories of damages considered (D, DPC and DPW) and precipitation (log-transformed) in Catalonia. Even if a linear regression indicates a significant link (p-value<0.01), the explanatory power of the model for D is rather low ($r^2$=0.09). Marginally better results are obtained for the damage indicators DPC and DPW ($r^2$=0.14 and $r^2$=0.16 respectively), underlying the importance of considering the impacts of population and wealth on damage. That is, this analysis corroborates the common experience that, given the same level of heavy precipitation, the total damage

is larger where the level of wealth is higher.

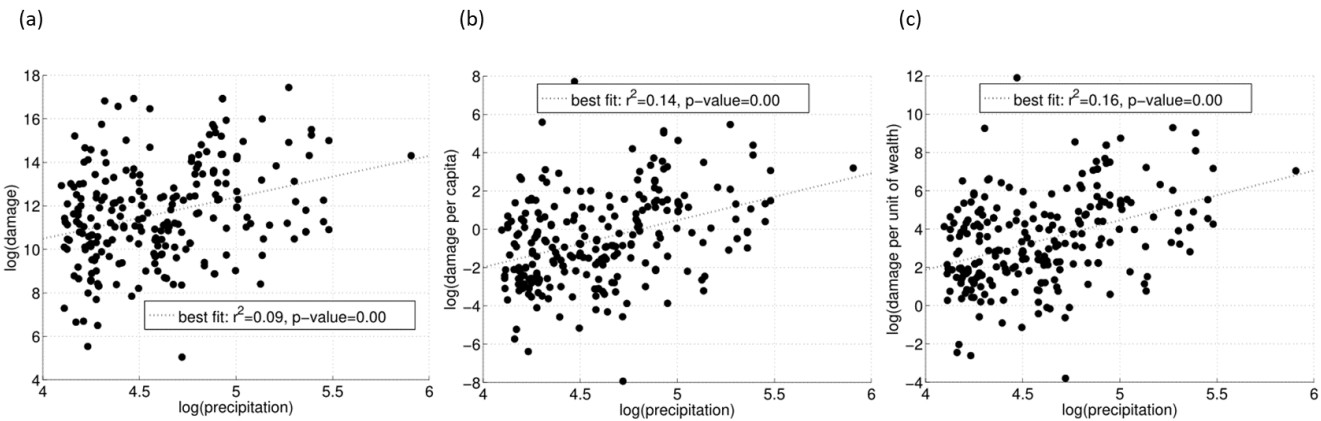

**Figure 2: Scatter plot showing maximum precipitation in 24 h (mm) and (a) total damages (D); (b) damage per capita (DPC); and (c) damage per unit of wealth (DPW), for flood events recorded in Catalonia between 1996 and 2015 (log-transformed values; damage**

**are given in euros). Each point represents the insurance compensation series (D, DPC or DPW) and the maximum 24 h precipitation for each basin. The dashed line indicates the fit based on a linear regression model.**

The large spread of Figure 2 indicates that modelling insurance compensations is a complex issue, due to the limitations in observational data and the concurrence of a variety of relevant factors. For instance, monetary data could be affected by

limitations, as the value of the assets exposed and insurance coverage may change over time (Barredo et al., 2012). Unfortunately, exact data on the value and location of assets exposed are not available.

However, the significant correlation between insurance compensations and precipitation suggests that rainfall data can be used to extract information on damage in Catalonia. To do this, we applied a logistic regression model to gauge the probability of large damaging events occurring given a certain precipitation amount (an approach that is frequently used for this kind of modelling study: Kim et al., 2012; Wobus et al., 2014). That is, our aim is not to estimate the precise amount of the monetary compensation, but to estimate when a "large" damaging event will occur given a certain precipitation amount. Since there is not a standard definition of a large damaging event, we tested several cases: insurance compensations exceeding the 50th, the 60th, the 70th, the 80th and the 90th percentile of the total sample. This methodology is repeated for both thresholds (40 mm and 60 mm) and for the three damage indicators (D, DPC, DPW) for the basins and warning areas. It means we made a total of 60 models.

Finally, the logistic model is calculated following the Eq. (1):

$$log(\frac{\pi}{1-\pi}) = \beta_0 + \beta_1 \cdot P, \tag{1}$$

where, $\pi$, are the response variable (i.e. the probability above a certain percentile) and P is the predictor (precipitation in our case). The value of the $\beta$ coefficient is determined using Generalized Linear Models (GLM). The Wald chi-square statistic is used to assess the statistical significance of individual regression coefficients (Harrel, 2015).

### 2.4 Verification method

We plotted the relative operating characteristic (ROC) diagram, a commonly-used logistic prediction diagnostic, showing the hit rate (i.e. the relative number of times a forecasted event actually occurred) against the false alarm rate (i.e. the relative number of times an event had been forecasted but did not actually happen) for different potential decision thresholds (Mason and Graham, 2002). Thus, for each insurance compensation percentile and for each precipitation threshold, we first calculated the forecast probabilities for that event, and then grouped the probability forecasts into batches (here 20 with a width of 0.05) to count the observed occurrences/non-occurrences. That is, we converted the observed and forecasted series, expressed as continuous amounts, into "exceedance" categories (yes-no statements indicating whether the data equals or exceeds selected probability). We then plotted the resulting elements on a standard contingency table (see Table 2).

The ROC diagram shows the Hit Rate (H) against the False Alarm Rate (F). These indices are defined as:

$$H = \frac{a}{a+c}; 0 \le H \le 1 \tag{2}$$

$$F = \frac{b}{b+d}; 0 \le F \le 1 \tag{3}$$

## 3 Results

### 3.1 Damaging events and precipitation in Catalonia

The total number of flood events recorded in Catalonia for the 1996-2015 period was 166 (109 of them went beyond the 40 mm/24 h precipitation threshold and 81 went over the 60 mm/24 h threshold) resulting in a total number of flood cases (i.e. pair of precipitation-damage values at a basin scale) of 642 (331 for 40 mm/24 h and 239 for 60 mm/24 h). Coastal municipalities are the most affected by flood events and where there is the most damage. This is a consequence of high vulnerability (the most vulnerable structures and infrastructures are on the coast), exposure (population and tourism are concentrated in the coastal regions) and hazards (floods associated to local heavy rain events are frequent) (Llasat et al., 2014, 2016a). Around 49 % of the events occurred during the months of July, August and September, with the latter month having the highest percentage of events (22 %). The most severe or catastrophic events occurred in the autumn, with 77 % of the events taking place between September and November (Llasat et al., 2016a). The compensations paid by the CCS for floods during this period in Catalonia amounted to €436.4 million.

Figure 3 shows the number of flood events recorded between 1996 and 2015 (Fig. 3a), the total insurance losses paid by CCS for flooding (Fig. 3b) during this period, the average population (Fig. 3c) and the GDP (Fig. 3d) in each basin. In general, there is a good correlation between the four variables, as expected. The basins with more recorded flood events are those that received more insurance compensations for flood damages, with a higher population and gross domestic product. The Maresme basin was affected by 41 % of the recorded events (Fig. 3a) with damages that add up to €24,561,762.4 between 1996 and 2015 (Fig. 3b).

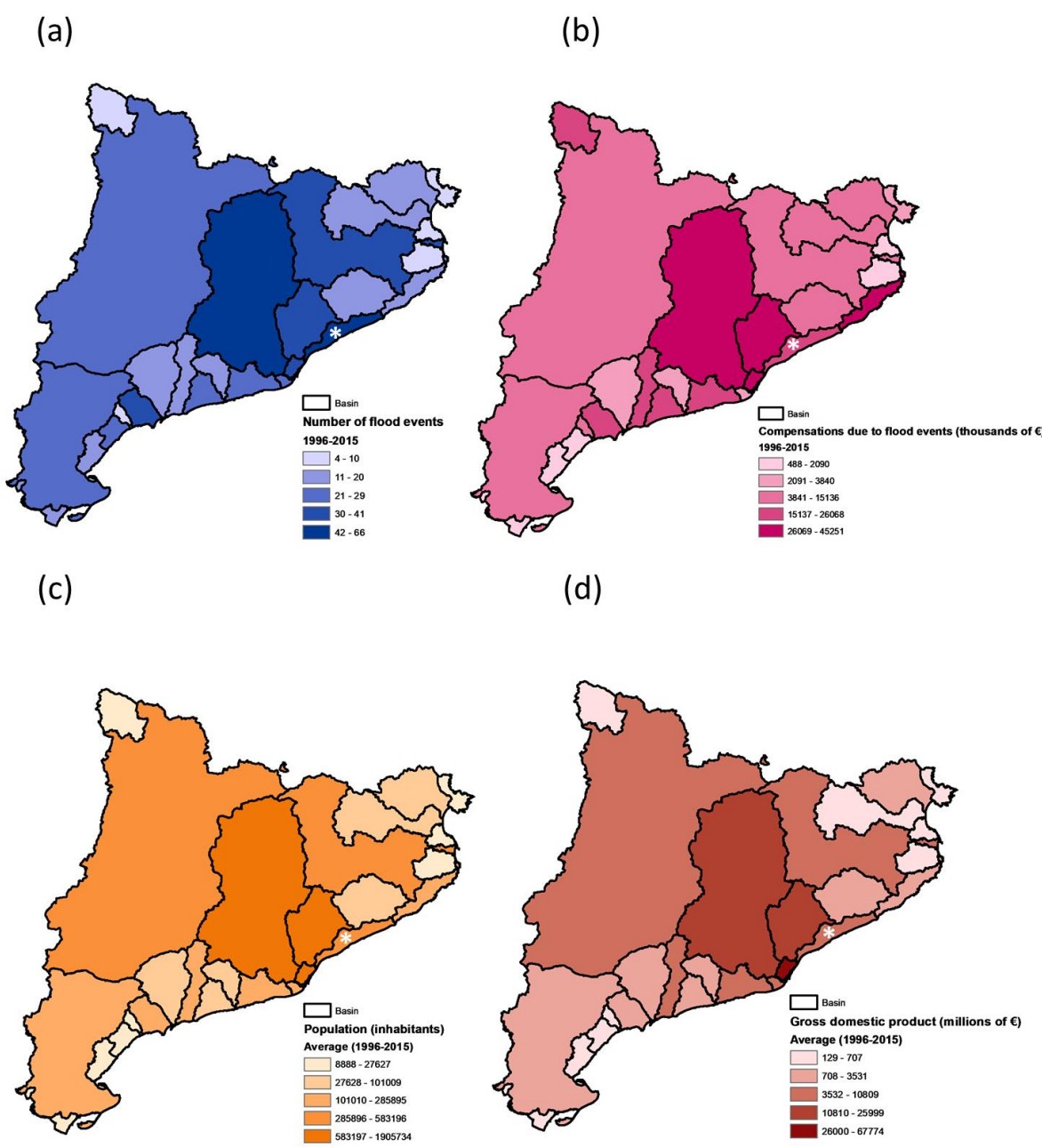

**Figure 3: Basin distribution of (a) flood events (1996-2015); (b) total insurance compensations for floods made by CCS (1996-2015); (c) average total population; and (d) average gross domestic product. Asterisk indicates Maresme basin.**

In order to estimate when a "large" damaging event will occur with a given precipitation amount, a logistic regression was used. Figure 4 shows a logistic regression example that indicates the model is able to simulate the probability of DPW above and below the 70th percentile as a function of precipitation. This figure illustrates that the probability of reaching above the 70th percentile for DPW increases when there is a large amount of rain. This result is consistent with the hypothesis that 24 h precipitation could be considered a good indicator for flood risk. For this example the regression equation [Eq. (4)] would be:

$$log\left(\frac{\pi}{1-\pi}\right) = -10.5 + 2.08 \cdot P, \tag{4}$$

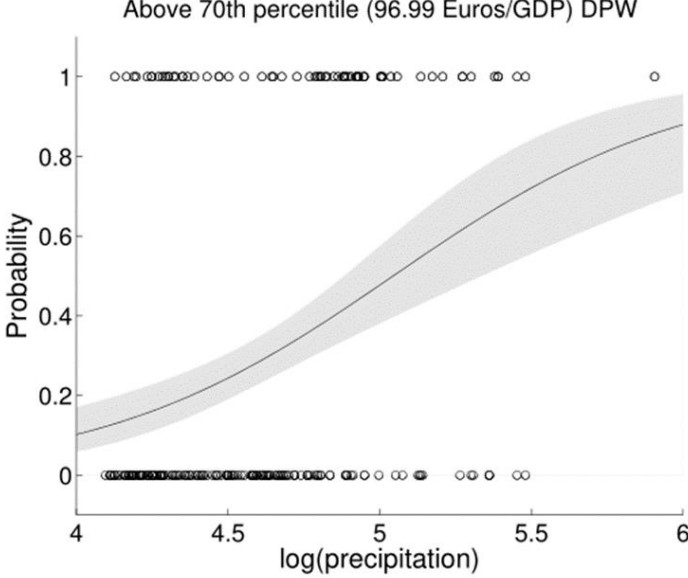

Figure 4: Example of logistic regression result to model DPW damages above the 70th percentile as a function of precipitation (log-transformed of the precipitation given in mm). The solid line indicates the best estimate while the shaded band indicates the 95 % confidence interval. Open circles along the horizontal axis show the events that are above (top) and below (bottom) the 70th percentile.

Table 3 shows the values of $\beta_0$ and $\beta_1$, considering cases with a threshold of 60 mm for the different combinations of damage indicators and percentiles.

It is important to assess whether this model can be used to separate positive and negative anomalies. Our models are not deterministic and users need to take into account the uncertainty of the forecast expressed by these probabilities. For example, users could decide to take action when a 10 % probability of an above-70th percentile event is forecast. In this case most of the observed events are forecasted, that is, the hit rate (i.e. the relative number of times a simulation event actually occurred) is close to 1, but this also implies a higher false alarm rate (i.e. the relative number of times an event had been simulated but did not actually happen). On the other hand, if a higher threshold is used, we can reduce the number of false alarms, but at the

expense of a greater number of missed events. The choice of the decision threshold is a function both of the skill of the forecast and the cost/loss ratio of the user. In any case, in a forecasting system affected by uncertainties, missed events can be reduced only by increasing false alarms and vice versa. In order to validate the model, we considered the ROC diagram (see Figure 5).

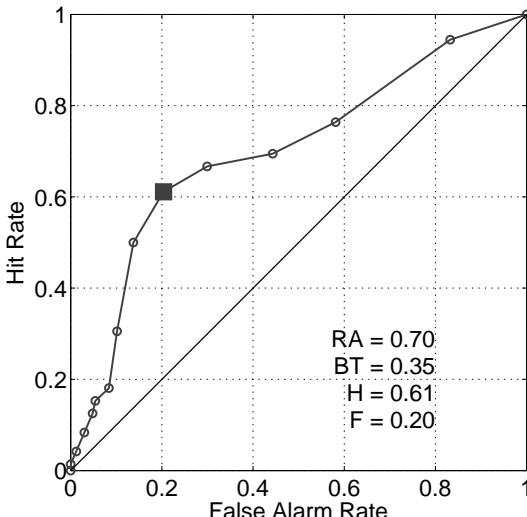

**Figure 5: Relative operating characteristic (ROC) diagram for above 70th DPW predictions using the logistic regression of Eq. (1). The open dots indicate a set of probability forecasts by stepping a decision threshold with 5 % probability through the modelling results. The numbers inside the plots are the ROC Area (RA) and the Best Threshold (BT), here defined as the threshold that**
**maximises the difference between the hit rate (H) and the false alarm rate (F).**

The area under the ROC curve (RA) is a useful measure to summarise the skill of a model. RA ranges from 0, for a forecast with no hit and only false alarms, to 1, indicating a perfect forecast. Models with an RA above 0.5 have more skill than random forecasts. Figure 5 shows that our model has skill: the ROC curve is well above the identity line, with an RA of 0.7. The "best
threshold" in this illustrative example is 0.35. This means that if we want to maximise the H-F difference (but please note that users could define other best thresholds according to their cost/loss ratio), an above 70th percentile damaging event is to be expected when our model predicts a probability higher then 0.35, resulting in H=0.61(this means that 61 out of 100 events are correctly modelled) and F=0.20 (this means that 20 out of 100 events were modelled as an "event" when it did not actually happen). For example, in this case (BT=0.35) a precipitation amount higher than 115 mm is needed to expect a damaging event
above the 70th percentile for the damage indicator DPW (97 euros/GDP).
Table 3 summarises the model parameters and performance considering all the percentiles and the three categories of damage used. In each case, precipitation is a significant predictor (p-value<0.05) and the models have skill and significant RA values (the significance is estimated using a Mann-Whitney U-test; Mason and Graham, 2002). Similar results were obtained for the

damage categories, with slightly larger RA considering DPW. Finally, a number of sensitivity checks were carried out. We repeated the analysis considering (i) a precipitation threshold of 40 mm instead of 60 mm, (ii) the AEMET warning areas, (iii) only coastal regions and (iv) the basin-averaged precipitation instead of the maximum values, obtaining similar results (see supplementary material).

### 3.2 Damaging events and precipitation in the Metropolitan Area of Barcelona

A total of 61 flood events were recorded in the Metropolitan Area of Barcelona (Fig. 1), which means an average of more than 3 events per year. The summer and autumn months had the highest number of flood events, with September having the most (31 %), followed by October (16 %). The insurance compensations paid by the CCS for floods amounted to € 86.3 million,

10 which represents 20 % of the total compensation paid by the CCS in Catalonia. The municipality of Barcelona recorded a total of 37 events between 1996 and 2015, all due to in situ precipitation and drainage problems in the city (Llasat et al., 2016b). The city of Barcelona also receives the most compensation for floods (around € 19 million).

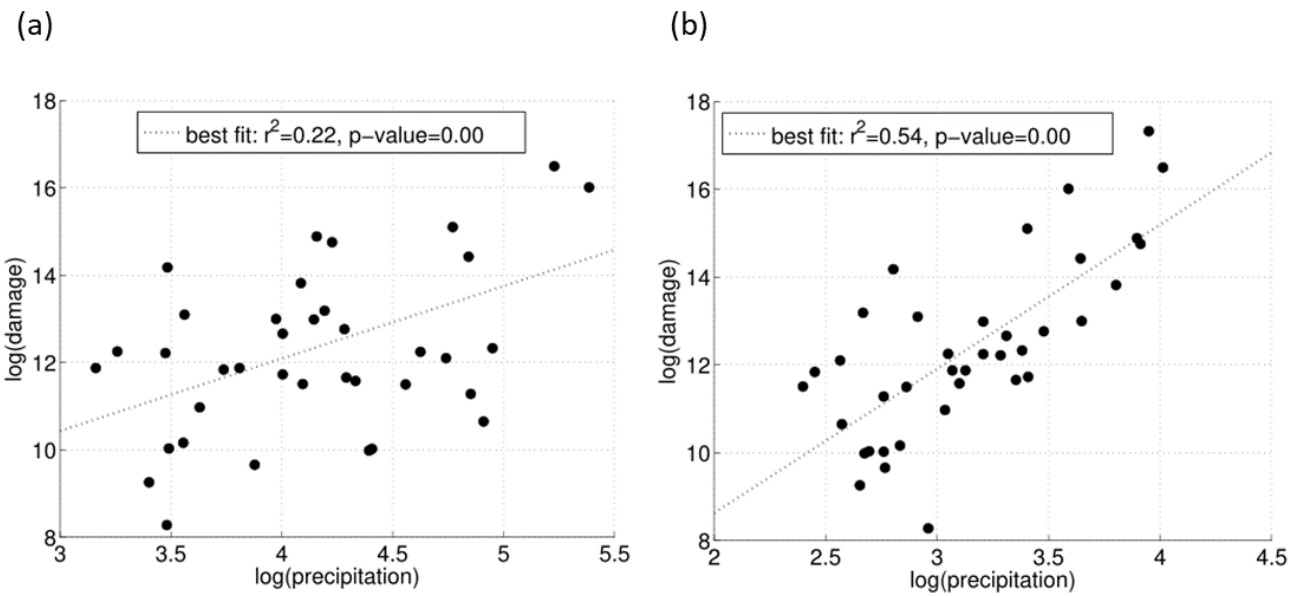

15   **Figure 6: Scatter plot (a) damages (D) versus 24 h precipitation and (b) damages (D) versus 30 minute precipitation (unit: log(mm)).**

As it can be seen in Figure 6, the total damages (D) relate more to 30 minute precipitation than to 24 h precipitation, with significant results in both cases. In this particular case, similar results are obtained for the other damage categories (DPC and DPW, see Table 4).

We then repeated the logistic modeling exercise using 30 minute precipitation. Figure 7 shows a logistic regression for the events that affected the MAB. As in the basin level aggregation, the model is capable of simulating the probability of total damage (D) above and below the 70th percentile as a function of 30 minute precipitation in this case. As could be expected, this probability increases with precipitation. The same methodology was applied using a precipitation threshold of 20 mm/30 min (see supplementary material) and using the 50th, 60th, 80th and 90th percentiles (Table 4). For this example, the regression equation would be:

$$log\left(\frac{\pi}{1-\pi}\right) = -11.3 + 3.21 \cdot P, \tag{5}$$

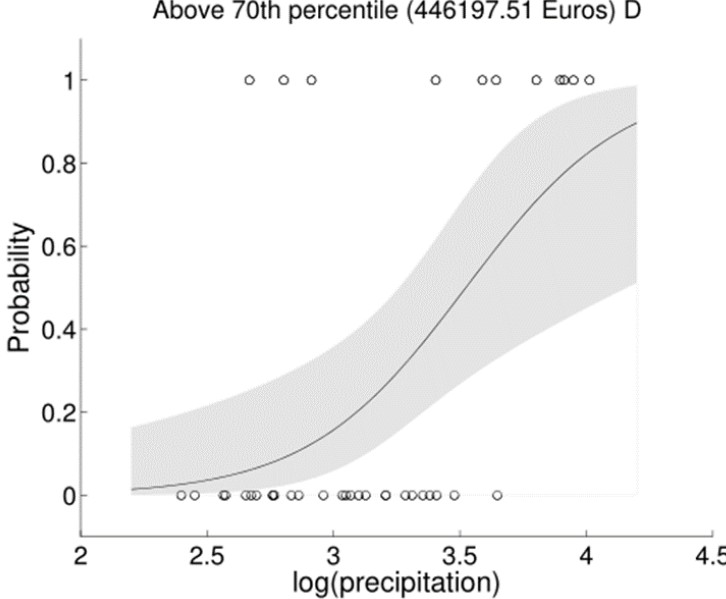

**Figure 7: Example of a logistic regression result to model damages (D) above the 70th percentile as a function of 30 minute precipitation (unit: log (mm)) for the MAB. The solid line indicates the best estimate while the shaded band indicates the 95 % confidence interval. Open circles along the horizontal axis show the events that are above (top) and below (bottom) the percentile 70th.**

Figure 8 shows the ROC diagram for predictions of total damages (D) above the 70th percentile for the MAB, using a precipitation threshold of 10 mm/30 min. The total RA (0.81) shows that our model for the MAB has skill. In this case, we would obtain the biggest difference between the hit and false rates when our model predicts a probability higher than 0.4. That is, the best threshold is 0.40, with 73 % of the events well-predicted (H=0.73) and only 11 % of false alarms events (F=0.11). In this example, a precipitation amount higher than 30 mm/30 min is needed to expect a damaging event above the 70th percentile for damage indicator D (0.45 million of euros).

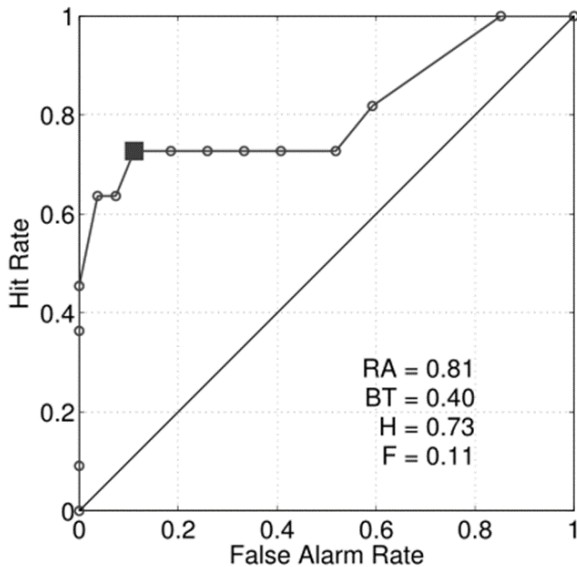

**Figure 8: Relative operating characteristic (ROC) diagram for predictions for damage indicator D above the 70th percentile for the MAB using the logistic regression of Eq. (1). The open dots indicate a set of probability forecasts by stepping a decision threshold with 5 % probability through the modelling results. The numbers inside the plots are the ROC Area (RA) and the Best Threshold (BT), here defined as the threshold that maximises the difference between the hit rate (H) and the false alarm rate (F).**

Table 4 summarises the model parameters and performance considering all the percentiles and the three damage categories used for a precipitation threshold of 10 mm/30min (see supplementary material for results using 20 mm/30 min for the MAB). Similar results in terms of RA have been obtained for damage categories, whether using a 10 mm (Table 4) or a 20 mm threshold (supplementary material).

## 4 Conclusions

The Mediterranean is an area affected by flood events that produce significant socioeconomic damage. Catalonia, located to the west of the Mediterranean, is affected by an average of more than 8 events per year. The majority of the damage caused by these events is due to local events, with intense and short-lived rainfall rather than river overflows (Llasat et al., 2014). Therefore, it is assumed that precipitation is the main contributing factor for damages caused by this type of event. To corroborate this hypothesis, the relationship between precipitation and compensation paid by insurance companies was studied. To take into account the differences in vulnerability and exposure in the territory, we considered three types of damage: total damage, damage per capita (divided by the population) and damage per unit of GDP.

Although linear regression indicates a significant link (p-value<0.01), suggesting that rainfall data can be used to extract information on damages in Catalonia, the variance explained for the model is rather low ($r^2$=0.09 for D, $r^2$=0.14 for DPC and $r^2$=0.16 for DPW). For this reason, the relationship was assessed using logistic regression models in order to estimate the probability of large monetary damages occurring as a result of heavy precipitation events. That is, our aim is not to estimate
the precise amount of insurance compensations, but to estimate when a "large" damaging event will occur given a particular precipitation amount. As could be expected, the logistic regression shows an increase in the probability of a damaging event occurring when precipitation increases. Our model is able to simulate the probability of a damaging event as a function of precipitation. In order to validate the model, we considered the Relative Operating Characteristic (ROC) diagram. The area under the ROC curve (RA) proved our model skill. The results show an RA above 0.6 in all percentiles of the three types of
damages and thresholds of precipitation, most of them with values higher than 0.7.

The methodology was also been applied for the MAB region, an urban area affected by more than three flood events per year. Linear regression has shown that 30 minute precipitation is linked more closely with damages than 24 h precipitation. That is, we repeated the analysis for 30 minute precipitation and, as expected, the model presents better results in terms of RA for the
urban area than for Catalonia as a whole, with values higher than 0.8 in all cases. Therefore, we have been able to confirm that 30 minute rainfall is a better predictor of the probability of large damages than daily rainfall in urban areas.

These results confirm the hypothesis that precipitation is a key factor in explaining the damage caused by flood events in regions in which water surface floods are the main type of floods, as is the case in the Mediterranean region of study. The
parsimonious empirical models linking flood damages to heavy precipitation are a step towards providing a substantial contribution to developing a warning forecast system with flood management strategies. For instance, from the relationship shown between precipitation and insurance compensations it is possible to predict when damaging events will occur as a result of a certain precipitation threshold. In other words, we have developed a new model that allows us to predict the probability that a flood event causing significant damages (where the meaning depends on the user) will occur, based on precipitation, and
taking into account the exposure and vulnerability of the region in the model.

These results were obtained by following a simple and transparent statistical methodology that can also be applied to other areas. These links could also provide a basis to predict flood damage in future climate change scenarios. It is worth noting that the complex relationships between climate variability, human activities and flood damages may limit the applicability of these findings to conditions that are very different from current ones. In addition, more complex analyses including more
sophisticated empirical methods, and other factors such as soil physical characteristics (e.g. slope, soil characteristics, vegetation), could provide additional understanding on flood drivers and impacts. Despite these limitations, this work has provided the first assessment of the link between precipitation and flood damages in a Mediterranean region, and our results

suggest that by exploiting the relationship between precipitation and flood damages, the model could provide a satisfactory prediction of monetary compensation.

**Competing interests**

The authors declare that they have no conflicts of interest.

**Acknowledgments**

This work has been supported by the Spanish Project HOPE (CGL2014-52571-R) of the Ministry of Economy, Industry and Competitiveness, the Metropolitan Area of Barcelona Project (no. 308321) (Flood evolution in the Metropolitan Area of Barcelona from a holistic perspective: past, present and future) and the Water Research Institute (IdRA) of the University of
Barcelona. It was conducted under the framework of the HyMeX Programme (HYdrological cycle in the Mediterranean EXperiment) and the Panta Rhei WG Changes in Flood Risk. We would like to thank AEMET and SMC for the meteorological and hydrological information provided for this study. Thanks also to BCASA for the detailed information about the system used to prevent and manage floods. Marco Turco was supported by the Spanish Juan de la Cierva Programme (IJCI-2015-26953). We would also like to acknowledge Hannah Bestow for correcting the English language of this paper.

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

**Tables**

Table 1: Summary of the data used. Precipitation refers to the number of meteorological stations considered; the number of flood events is the total sum for the period 1996-2015; the average population is the total number of inhabitants; the average Gross Domestic Product is in millions of euros; the damages refer to the compensations paid by the CCS for the 1996-2015 period in millions of euros.

| 1996-2015 | CATALONIA | MAB | SOURCE |
|---|---|---|---|
| Precipitation 24 h | 127 | 26 | AEMET |
| Precipitation 30 minute | - | 14 | SMC |
| Number of flood events | 166 | 61 | INUNGAMA/PRESSGAMA |
| Population | 6,854,302 | 3,141,703 | IDESCAT |
| Gross Domestic Product | 164,162.3 | 95,438.57 | IDESCAT |
| No. of municipalities | 948 | 36 | IDESCAT |
| Damages | 436.4 | 86.3 | CCS |

Table 2: Contingency table to support Equation 2 and 3.

| | | OBSERVED | |
|---|---|---|---|
| | | Yes | No |
| FORECASTED | Yes | a | b |
| | No | c | d |

25

Table 3: Parameters of the logistic model and RA values for the basin level with 60 mm/24 h maximum precipitation threshold. All the results are significant (p-value<0.01). Number of flood cases: 239.

| PERCENTILE | DAMAGE | $\beta_0$ | $\beta_1$ | RA |
|---|---|---|---|---|
| 50 | D | -5.31 | 1.16 | 0.61 |
|  | DPC | -9.19 | 2.00 | 0.67 |
|  | DPW | -8.73 | 1.90 | 0.67 |
| 60 | D | -6.89 | 1.41 | 0.64 |
|  | DPC | -8.90 | 1.84 | 0.67 |
|  | DPW | -9.58 | 1.99 | 0.68 |
| 70 | D | -7.65 | 1.47 | 0.65 |
|  | DPC | -11.26 | 2.24 | 0.72 |
|  | DPW | -10.50 | 2.08 | 0.70 |
| 80 | D | -10.19 | 1.89 | 0.70 |
|  | DPC | -10.44 | 1.94 | 0.70 |
|  | DPW | -11.84 | 2.24 | 0.73 |
| 90 | D | -11.13 | 1.90 | 0.71 |
|  | DPC | -11.58 | 1.99 | 0.70 |
|  | DPW | -12.86 | 2.26 | 0.74 |

Table 4: Parameters of the logistic model and RA values for the MAB level with 10 mm/30 minute maximum precipitation threshold. All the results are significant (p-value<0.05). Number of flood cases: 38

| PERCENTILE | DAMAGE | $\beta_0$ | $\beta_1$ | RA |
|---|---|---|---|---|
| 50 | D | -14.61 | 4.7 | 0.88 |
|  | DPC | -14.61 | 4.7 | 0.88 |
|  | DPW | -10.02 | 3.21 | 0.81 |
| 60 | D | -13.34 | 4.06 | 0.85 |
|  | DPC | -13.34 | 4.06 | 0.85 |
|  | DPW | -13.72 | 4.18 | 0.86 |
| 70 | D | -11.30 | 3.21 | 0.81 |
|  | DPC | -11.30 | 3.21 | 0.81 |
|  | DPW | -15.05 | 4.33 | 0.87 |
| 80 | D | -16.62 | 4.58 | 0.89 |
|  | DPC | -16.62 | 4.58 | 0.89 |
|  | DPW | -16.62 | 4.58 | 0.89 |
| 90 | D | -17.72 | 4.53 | 0.91 |
|  | DPC | -17.72 | 4.53 | 0.91 |
|  | DPW | -17.72 | 4.53 | 0.91 |