# Peer review of "The relationship between precipitation and insurance data for floods in a Mediterranean region (Northeast Spain)"

_Natural Hazards and Earth System Sciences, 2017_

## Referee Comment (RC1) · Anonymous Referee #1 · 24 Aug 2017

General comments

The manuscript is analysing the link between the causes and impacts of floods by means of precipitation measurements and insurance claims. The main objective of this study is to identify the best indicators for describing this relationship. The topic is of great interest.

However, the manuscripts is weak due to a few but important points.

The use of the Spearman rank test on the data is to test the correlation between two indicators. However, this test is not providing any information for drawing conclusion. The correlation between precipitation and flood losses is to be expected and reported

anywhere. Indeed, other literature goes far behind correlation only and analyses the form of the relationship resp. provides e.g. minimal rainfall intensity thresholds for losses. The added value of this study is therefore disputable.

A detailed definition of the terms "floods", "pluvial floods", "flash floods" and "urban floods" is missing. It is not clearly defined which processes are relevant for the respective insurance claims. This is important because of the chosen approach in defining the aggregation units. Depending on the size of the aggregation unit, the spatial distances between rainfall (in the catchment) and flood impacts (in the floodplain) may be very different for riverine floods and pluvial floods (in situ precipitation).

The authors chose three different spatial aggregation units: regional, basin, and local scale. With this, the study ignores the Modifiable Areal Unit Problem (MAUP) as described in Openshaw (1984). The size and shape of the aggregation units may influence the results of the test. This arises because the authors do not explicitly differentiate between losses to houses due to pluvial floods and riverine floods as exemplarily shown by Bernet et al. in this special issue (https://doi.org/10.5194/nhess-2017-136).

The interpolation of the rainfall measurements of the single stations and the aggregation method to the different spatial units is not described.

The authors analyse insurance claims in the period 1996-2015. While the compensations are adjusted with the consumer price index, the increase in the total stock resp. changes in the overall exposure to potentially flooded areas in this period are not considered. In the study period, a relatively high increase in the total building stock should be expected due to the construction activities before the financial crisis in 2008. Thus, the losses in the insurance dataset may be supposed to a remarkable instationarity.

The analysis of the correlation between precipitation and compensation paid was made on the basis of the recorded flood events (or flood episodes). The definition of a flood event (above 75th percentile) is not made transparent resp. not clearly enough described. The paper has to be reworked fundamentally.

References: Openshaw, Stan, 1984. TheModifiable Areal Unit Problem. Geo Abstracts Univ. of East Anglia, Norwich (Catmog: concepts and techniques in modern geography).

---

## Referee Comment (RC2) · Anonymous Referee #2 · 18 Sep 2017

**1 Summary of the article**

The authors establish a linear relationship between the logarithm of the triggering rainfall and the logarithm of the resulting insurance claims of flash floods in the region of Catalonia (Spain). The significance and magnitude of the correlation coefficient is used as the main argument to proof the hypothesis that precipitation alone is sufficient to predict the magnitude of insurance claims resulting from flash floods in that very region.

**2 Review**

After this summary lets jump into the study itself. I will try to describe the work with my own words, stopping here and there to add my toughts and concerns.

**2.1 Data**

Three prinicipal sources of information are used by the authors. First the Inungama database from which basic data about flash floods in Catalonia are drawn. These are:

- affected municipalities
- · affected basins
- start and end date of the event.

An event in the context of this article is therefore, as I understood it, an entry in the Inungama database.

At this point the authors know when a (flash) flood happend and which municpialies in which basin were affected. Now the second source of data is entering the stage: the flood damage data from the Spanish Insurance Compensation Consortium (CCS). The event data is at the municipality level and therefore the CCS is aggregated also at that level (which is the finest grain of spatial resolution for this study). By performing a join based on the smallest temporal distance between the event and the date of the insurance claim every event should now also have a variable called *Compensations*.

The last data source is meteorological data from the Spanish State Meteorological Agency. This should add another collumn to the data set with the accumulated 24h precipitation on the event day.
**Suggestion 1:** Because the focus of the article is on flash floods the authors should only include flash floods in their analysis. The difference between flash and non-flash floods must then be stated clearly maybe a (working) definition of flash floods could be based on the length of the event (less than 24 hours).

**Suggestion 2:** Figure 4 is showing the number of floods and the amount of compensation per municipality. Some of the municipalities which have no flood event have compensation payments suggesting a flaw in the homogenisation procedure or simply a graphical one because the legend of figure 4b starts at 0 with a light pink tone.

**2.2 Aggregation**

If I got it straight the dataset should consit of entries with the following structure: a flash flood event *i* affected  $n_i$  municipalities in  $m_i$  basins. From the  $n_i$  affected municipalities  $n_i - k_i$  received compensations of  $Y_i$ . The anticipated cause of the event is the 24h precipitation  $X_{i,j}$  recorded at the day of the event at station *j*. Next the authors try to find the pair  $(Y_i, X_{i,j})$  which yields the highest correlation in the log-log plane.

Let us, for the moment, assume that the hypothesis: payed compensation is a linear function of precipitation,

$$\log(Y) = a + b * \log(X)$$

is true. How could this be physically possible? First the compensation payed to cover the damage is caused by a flood event. The flood is produced by a stream (may it ephemeral or not) and this stream has a basin. Finnally the precipitation collected by the basin is the fuel for the catastrophic machinery producing the flood. It follows that only the amount of precipitation in the basin of the damage causing-stream should be related to the amount of compensation. Sounds logical tome. The authors find that the maximum precipitation over all affected basins has the highest correlation with the sum of compensations in the affected basins. This is a minor contradiction with the flow of reasoning presented which I assume also the authors used. But further problems NHESSD
may emerge like that the damage itself depends on the number of damageable objects (exposure) in the basin aka at the time of the event. Let us assume that a rainfall of  $P_x$  is causing the total damage of a building in a basins of size  $A_x$  for all buildings with a distance to the stream of say  $d_x$  then only changing the number of building in the buffer  $d_x$  will result in considerable difference in the amount of compensation.

**Suggestion 3:** The exposure should be taking into account in other words a relative compensation should be formulated as the response variable in the analysis.

**Suggestion 4:** Adding a scatterplot of precipitation versus compensations for all used aggregation procedures would strongly enhance the understanding of the results.

**2.3 Results**

The authors present with figure 5 the key results of the regional analysis. Only guessing from the figure a linear model should be seriously influenced by the obersavtion at x = -1. If the log is the logarithm to the base 10 than this is a precipitation value of 0.1 mm which also seems unrealsitic. The authors also state a precipitation threshold (100 mm) at which significant damages are observed suggesting that the probability of having a damage above 30.000 is maximized if the precipitation is above 100 mm no further explanation nor quantification is given.

**Suggestion 5:** Look at the observation with the low precipitation in more detail. Is it a measurement error? Maybe their is a wrong decimal sign? Is it really a flash flood and is it caused by precipitation? Generally the definition of the analysed data should be made more precise aka the obersavtions should be checked if they belong to the set of interest aka not comparing apples with oranges.

The analysis on the basin scale is focusing on a black and white example: a basin showing high correlation and therefore supporting the hypothesis of the authors and on the other hand a basin with low correlation contradicting the hypothesis (the mean NHESSD
correlation for all basins is 0.47 (se  $\pm$  0.4) which is rather low). To resolve the low correlation in the black basin the authors split the data set according to a population by maximizing the correlation coefficient turning the black into a white one.

**Suggestion 6:** Using the population as a basis for classifing rural and urban regions reminds me of using a dummy variable in regression from their it is only a slight jump to use population as variable in conjunction with precipitation. Using a ANOVA (or testing against a 0 slope of population or precipitation) would do the trick to see which one of the two is more important. But following suggestion 3 the influence of the population should vanish if and only if the hypothesis of a linear model in only influenced by precipitation is correct.

The last subset of observation is the MAB (metropolian area of barcelona) suggesting that a finer temporal grain (30 min) of the precipitation is enhancing the predtiction of compensation payments. Then the precipitation is correlated with the precipitation in 24h which results in a low correlation. Now the whole other data analysis is based on the 24h precipitation but the 30 min seems to be better suited. What are the implications for the 24h precipitation used for the other data sets?

**Suggestion 7:** Presenting scatter plots are much better suited than maps in my humble opinion. The whole point of the study is the assumption of linearity between precipitation and compensation and simple plot could demonstrade this with elegant ease.

**3 Final Statement**

I hope the review was not unpolite and has in any way offented the auhtors which was not at all my purpose. I think the study needs a major overhaul regarding the data preprocessing as well as the techniques used to draw conclusions.

**NHESSD**

---

## Referee Comment (RC3) · Anonymous Referee #3 · 6 Oct 2017

**Article summary**

The article analyses the correlation between extreme rainfall events and compensation costs triggered by flash floods, which are drawn from insurance records. The correlation coefficient is used to draw conclusions on the causal effect between precipitation and damage magnitude, using different scales of aggregation as tests.

**General comments**

The topic is of great importance and the use of empirical data is a plus, however the thinking behind the paper is a bit too much straightforward. Indeed precipitation is a major driver of flash flood damage; but it is not the only factor. The paper do not
take in account other factors influencing damage (slopes, land cover and soil sealing, vegetation), and explains the effect (damage) by stating the cause (heavy rainfall); the conclusion uses correlation values to confirm the hypotesis.

The statistical analysis needs to go deeper and to add more insights in relation to the distribution of damage along different typologies of exposure. The analysis uses 4 different aggregation scales based on admistrative units; most commonly in these kind of studies the scale would be smaller than the municipality. A projection of the data over built-up areas from land cover, building units or a regular grid cells would improve the analysis by linking the variables at a more detailed and homogeneus unit compared to the administrative boundaries. I would suggest then to present only the results relative to the better performing aggregation method, as the comparison on administrative units do not produce added value for the conclusions.

The difference between different kind of floods and to which kind exactly the compensatory records refer is not clearly stated in the paper. Overall, both the record data and the spatial data needs to be presented more precisely and clearly.

Maps can be easily reduced in numbers and made more readable: figure 1 "a" and "b" can be combined by showing only the necessary information (river, basins, population, scores). Same goes for figure 3, it could be combined into 1 or 2 showing the information (dots) in different shapes/colors.

Finally, I agree with the insightful comments by reviewer 1 and 2 and suggest to majorly revise the paper by rethinking its objectives and methods.

NHESSD

---

## Author Comment (AC1) · 24 Nov 2017

Manuscript: nhess-2017-278 "The relationship between precipitation and insurance data for surface water floods in a Mediterranean region (Northeast Spain)".

Responses to reviewer #1:

Reviewer #1 (Highlight): The manuscript is analysing the link between the causes and impacts of floods by means of precipitation measurements and insurance claims. The main objective of this study is to identify the best indicators for describing this relationship. The topic is of great interest. However, the manuscripts is weak due to a
few but important points.

Response: We wish to thank the anonymous referee for his/her useful and constructive comments. Each specific point has been addressed in the manuscript as explained in the following document.

Referee's Comment: The use of the Spearman rank test on the data is to test the correlation between two indicators. However, this test is not providing any information for drawing conclusion. The correlation between precipitation and flood losses is to be expected and reported anywhere. Indeed, other literature goes far behind correlation only and analyses the form of the relationship resp. provides e.g. minimal rainfall intensity thresholds for losses. The added value of this study is therefore disputable.

Response: We would like to thank you for this very important comment. To address it, we applied a logistic regression model to gauge the probability of large damaging events given a certain precipitation amount, an approach that is frequently used for this kind of modelling study (Kim et al., 2012; Wobus et al., 2014).

Referee's Comment: A detailed definition of the terms "floods", "pluvial floods", "flash floods" and "urban floods" is missing. It is not clearly defined which processes are relevant for the respective insurance claims. This is important because of the chosen approach in defining the aggregation units. Depending on the size of the aggregation unit, the spatial distances between rainfall (in the catchment) and flood impacts (in the floodplain) may be very different for riverine floods and pluvial floods (in situ precipitation).

Response: The insurance company (CCS) offers insurance compensations for all the claims related to flood damages, regardless of the type of flood. Nevertheless, the floods that most frequently affect the region of study, the Mediterranean, are caused by in situ precipitation (pluvial floods or, in broader terms, "surface water floods") (Llasat et al., 2014). Surface water floods can be thought of as the most general form of rainfall-related (pluvial) floods (Bernet et al., 2017). For this reason, the hypothesis of
the study is that precipitation is the main cause of damaging floods in Catalonia, and it is expected that flood insurance data will show a strong correlation (Cortès et al., 2017). Taking into account the limitations of the data, we decided to work on a basin scale. For each basin and event, the maximum 24 h precipitation and total flood compensation was estimated. In the case of the MAB, the availability of precipitation data for lower time intervals (30 minutes) allowed us to make a comparison for both results. Finally, we added this sentence in the manuscript: "Most floods that have affected the region of study, Northeast Spain, are surface water floods. This type of floods can be regarded as coming under the most general definition of rainfall-related floods (Bernet et al., 2017), including pluvial floods but also flooding from sewer systems, small open channels, diverted watercourses or flooding from groundwater springs (Falconer et al., 2009). River floods that affect great distances are very rare in the region, and are only related to catastrophic and extended floods (for the analysed period only the October 2000 floods were of this type). Nevertheless, these are usually absorbed by reservoirs. It is therefore expected that flood insurance data will correlate strongly with precipitation and surface water floods."

Referee's Comment: The authors chose three different spatial aggregation units: regional, basin, and local scale. With this, the study ignores the Modifiable Areal Unit Problem (MAUP) as described in Openshaw (1984). The size and shape of the aggregation units may influence the results of the test. This arises because the authors do not explicitly differentiate between losses to houses due to pluvial floods and riverine floods as exemplarily shown by Bernet et al. in this special issue (https://doi.org/10.5194/nhess-2017-136).

Response: As mentioned before, we decided to work by aggregating the data on a basin level and for the MAB region, because the available data are too sparse to support our statistical assessment on a municipal scale. On the other hand, the floods in question, as we have explained before, are pluvial floods. Furthermore, we considered another spatial aggregation based on the Spanish State Meteorological Agency
(AEMET) warning areas, obtaining similar results (included in the supplementary material).

Referee's Comment: The interpolation of the rainfall measurements of the single stations and the aggregation method to the different spatial units is not described.

Response: Due to the existence of few meteorological stations, for precipitation data we used the maximum over 24 h recorded for each basin and event. There was no interpolation process. In order to better clarify this point, we add: "Because the available data are too sparse to support our statistical assessment on a municipal scale, we assessed the precipitation-compensation link for Catalonia as a whole. That is, we sampled pairs of the response variable (i.e. the compensation series) and the maximum 24 h precipitation for each basin, and pooled them into one sample for the entire region (Catalonia) to correlate them. For each event there can be more than one pair of values, depending on the number of affected catchments. From now on we will use the expression "flood case" for each pair of values corresponding to a basin affected by a flood event. This area is large enough to have a fairly large sample size for analysis, but small enough that the causes of flood damages are likely to be similar across the area."

Referee's Comment: The authors analyse insurance claims in the period 1996-2015. While the compensations are adjusted with the consumer price index, the increase in the total stock resp. changes in the overall exposure to potentially flooded areas in this period are not considered. In the study period, a relatively high increase in the total building stock should be expected due to the construction activities before the financial crisis in 2008. Thus, the losses in the insurance dataset may be supposed to a remarkable instationarity.

Response: We agree with the reviewer. Modelling insurance compensations is a complex issue due to the limitations in observational data and the concurrence of a variety of factors that affects them. For instance, in order to statistically link rainfall and insurInteractive comment
ance compensation, both precipitation and monetary data would need to be compiled precisely and consistently over time and across the region. While precipitation data follows a formal quality control, the data for insurance compensations are no standardised. For instance, the value of assets exposed and insurance coverage may change over time (Barredo et al., 2012). Unfortunately, precise data on the value and location of assets exposed are not available. However, we show that rainfall data can be used to extract information on damages in Catalonia. To do so, we applied a logistic regression model to gauge the probability of large damaging events given a certain precipitation amount. That is, our aim is not to estimate the precise magnitude of the monetary compensation, but to estimate when a "large" damaging event will occur given a certain precipitation amount. In addition, we take into account the relative impacts of socio-economic factors on damage, considering not only the total damage, but also the damage per capita (DPC) and damage per unit of gross domestic product (DPW).

Referee's Comment: The analysis of the correlation between precipitation and compensation paid was made on the basis of the recorded flood events (or flood episodes). The definition of a flood event (above 75th percentile) is not made transparent resp. not clearly enough described.

Response: We have changed this part of the study. In order to define a flood event we used the INUNGAMA database (Barnolas and Llasat 2007; Llasat et al., 2016a), from which we took the event dates and duration. This database records all the flood events that have affected Catalonia, most of them caused by in situ precipitation (surface water floods).

Referee's Comment: The paper has to be reworked fundamentally.

Response: The revised manuscript has been thoroughly rewritten. One of the most important points is that we now model the compensations with a logistic regression strategy, testing the sensitivity of our results to the different threshold used to define the events. We have carried out a major overhaul of the data processing and the
methods and techniques applied. We are confident that these major changes have improved the statistical significance of the analyses, and improved the clarity of the results presented.

References:

Barnolas, M. and Llasat, M. C.: System Sciences A flood geodatabase and its climatological applications: the case of Catalonia for the last century, (2005), 271–281, 2007.

Bernet, D. B., Prasuhn, V. and Weingartner, R.: Surface water floods in Switzerland: What insurance claim records tell us about the damage in space and time, Nat. Hazards Earth Syst. Sci., 17(9), 1659–1682, doi:10.5194/nhess-17-1659-2017, 2017.

Cortès, M., Llasat, M. C., Gilabert, J., Llasat-Botija, M., Turco, M., Marcos, R., Martín Vide, J. P. and Falcón, L.: Towards a better understanding of the evolution of the flood risk in Mediterranean urban areas: the case of Barcelona, Nat. Hazards, 1–22, doi:10.1007/s11069-017-3014-0, 2017.

Falconer, R. H., Cobby, D., Smyth, P., Astle, G., Dent, J. and Golding, B.: Pluvial flooding: New approaches in flood warning, mapping and risk management, J. Flood Risk Manag., 2(3), 198–208, doi:10.1111/j.1753-318X.2009.01034.x, 2009.

Kim, Y. O., Seo, S. B., & Jang, O. J. (2012). Flood risk assessment using regional regression analysis. Natural hazards, 63(2), 1203-1217.

Llasat, M. C., Marcos, R., Llasat-Botija, M., Gilabert, J., Turco, M. and Quintana-Seguí, P.: Flash flood evolution in North-Western Mediterranean, Atmos. Res., 149, 230–243, doi:10.1016/j.atmosres.2014.05.024, 2014.

Llasat, M. C., Marcos, R., Turco, M., Gilabert, J. and Llasat-Botija, M.: Trends in flash flood events versus convective precipitation in the Mediterranean region: The case of Catalonia, J. Hydrol., 541 (September 2002), 24–37, doi:10.1016/j.jhydrol.2016.05.040, 2016a.

NHESSD
Openshaw, Stan, 1984. The Modifiable Areal Unit Problem. Geo Abstracts Univ. of East Anglia, Norwich (Catmog: concepts and techniques in modern geography).

Wobus, C., Lawson, M., Jones, R., Smith, J., & Martinich, J. (2014). Estimating monetary damages from flooding in the United States under a changing climate. Journal of Flood Risk Management, 7(3), 217-229.

Please also note the supplement to this comment: https://www.nat-hazards-earth-syst-sci-discuss.net/nhess-2017-278/nhess-2017-278-AC1-supplement.zip
**Fig. 1.** Map of Catalonia showing the aggregated basins, the Metropolitan Area of Barcelona (MAB), the main rivers and the pluviometric stations used.

---

## Author Comment (AC2) · 24 Nov 2017

Manuscript: nhess-2017-278 "The relationship between precipitation and insurance data for surface water floods in a Mediterranean region (Northeast Spain)".

Responses to reviewer #2:

Reviewer #2 (Summary of the article): The manuscript is analysing the link between the causes and impacts of floods by means of precipitation measurements and insurance claims. The main objective of this study is to identify the best indicators for describing this relationship. The topic is of great interest. However, the manuscripts is weak due

**Printer-friendly version**

to a few but important points.

Response: We would like to thank reviewer for her/his very constructive comments.

Referee's Comment: After this summary lets jump into the study itself. I will try to describe the work with my own words, stopping here and there to add my toughts and concerns. 2.1 Data Three principal sources of information are used by the authors. First the Inungama database from which basic data about flash floods in Catalonia are drawn. These are: aĂć affected municipalities aĂć affected basins aĂć start and end date of the event. An event in the context of this article is therefore, as I understood it. an entry in the Inungama database. At this point the authors know when a (flash) flood happend and which municipalies in which basin were affected. Now the second source of data is entering the stage: the flood damage data from the Spanish Insurance Compensation Consortium (CCS). The event data is at the municipality level and therefore the CCS is aggregated also at that level (which is the finest grain of spatial resolution for this study). By performing a join based on the smallest temporal distance between the event and the date of the insurance claim every event should now also have a variable called Compensations. The last data source is meteorological data from the Spanish State Meteorological Agency. This should add another collumn to the data set with the accumulated 24h precipitation on the event day. Suggestion 1: Because the focus of the article is on flash floods the authors should only include flash floods in their analysis. The difference between flash and non-flash floods must then be stated clearly maybe a (working) definition of flash floods could be based on the length of the event (less than 24 hours). Suggestion 2: Figure 4 is showing the number of floods and the amount of compensation per municipality. Some of the municipalities which have no flood event have compensation payments suggesting a flaw in the homogenisation procedure or simply a graphical one because the legend of figure 4b starts at 0 with a light pink tone.

Response: We wish to thank the anonymous reviewer for the description of the data, which allowed us to improve the explanation of our pre-processing of the data. In
the region of study (Mediterranean area) most floods are due to in situ precipitation (surface water floods). For this reason, our hypothesis is that precipitation is the main cause of damaging floods. Most of them are flash floods, events that last less than 24 hours, however this is not true in every case, and for this reason we worked on all the flood events recorded in the INUNGAMA database. In the manuscript we clarified this in the following sentence: "Most floods that have affected the region of study, Northeast Spain, are surface water floods. This type of floods can be regarded as coming under the most general definition of rainfall-related floods (Bernet et al., 2017), including pluvial floods but also flooding from sewer systems, small open channels, diverted watercourses or flooding from groundwater springs (Falconer et al., 2009). River floods that affect great distances are very rare in the region, and are only related to catastrophic and extended floods (for the analysed period only the October 2000 floods were of this type). Nevertheless, these are usually absorbed by reservoirs. It is therefore expected that flood insurance data will correlate strongly with precipitation and surface water floods." In the revised manuscript we work on a basin level. This domain is large enough to have a fairly large sample size for analysis (we select a total of 221 "cases"), but small enough that the causes of flood damages are likely to be similar across the area. We also focus on the MAB area, where higher resolution precipitation data are available. In addition, working at a higher level of aggregation allows us not only to reduce possible heterogeneities between the databases, but also to ensure more robust data for each unit.

Referee's Comment: 2.2 Aggregation. If I got it straight the dataset should consit of entries with the following structure: a flash flood event i affected ni municipalities in mi basins. From the ni affected municipalities ni – ki received compensations of Yi. The anticipated cause of the event is the 24h precipitation Xi,j recorded at the day of the event at station j. Next the auhtors try to find the pair (Yi, Xi,j) which yields the highest correlation in the log-log plane. Let us, for the moment, assume that the hypothesis: payed compensation is a linear function of precipitation,  $log(Y) = a + b \ aLU \ log(X)$  is true. How could this be physically possible? First the compensation payed to cover
the damage is caused by a flood event. The flood is produced by a stream (may it ephemeral or not) and this stream has a basin. Finnally the precipitation collected by the basin is the fuel for the catastrophic machinery producing the flood. It follows that only the amount of precipitation in the basin of the damage causing-stream should be related to the amount of compensation. Sounds logical tome. The authors find that the maximum precipitation over all affected basins has the highest correlation with the sum of compensations in the affected basins. This is a minor contradiction with the flow of reasoning presented which I assume also the authors used. But further problems may emerge like that the damage itself depends on the number of damageable objects (exposure) in the basin aka at the time of the event. Let us assume that a rainfall of Px is causing the total damage of a building in a basins of size Ax for all buildings with a distance to the stream of say dx then only changing the number of building in the buffer dx will result in considerable difference in the amount of compensation. Suggestion 3: The exposure should be taking into account in other words a relative compensation should be formulated as the response variable in the analysis. Suggestion 4: Adding a scatterplot of precipitation versus compensations for all used aggregation procedures would strongly enhance the understanding of the results.

Response: We would like to thank the reviewer again for the useful suggestion to improve our study. Taking this into account, we have completely reformulated our study. First of all, as the reviewer proposes, we completely agree with the need to include exposure in the data, and, for this reason, we have tested our model using not only absolute damage (D in the manuscript), but also damage per capita (DPC) and damage per unit of gross domestic product (DPW). This means the relative impacts of socio-economic factors on damage can be estimated while taking into account population and wealth responses. Taking into account suggestion 4, we have changed our figures, adding scatter plots for both levels of aggregation (basin and MAB) in order to show the relationship between precipitation and insurance data. In addition, we have added more graphs in the supplementary material with the different thresholds of precipitation used and also considering the Spanish State Meteorological Agency (AEMET) warning
**areas.**

Referee's Comment: 2.3 Results The authors present with figure 5 the key results of the regional analysis. Only guessing from the figure a linear model should be seriously influenced by the obersavtion at x = -1. If the log is the logarithm to the base 10 than this is a precipitation value of 0.1 mm which also seems unrealsitic. The authors also state a precipitation threshold (100 mm) at which significant damages are observed suggesting that the probability of having a damage above 30.000 is maximized if the precipitation is above 100 mm no further explanation nor quantification is given. Suggestion 5: Look at the observation with the low precipitation in more detail. Is it a measurement error? Maybe their is a wrong decimal sign? Is it really a flash flood and is it caused by precipitation? Generally the definition of the analysed data should be made more precise aka the obersavtions should be checked if they belong to the set of interest aka not comparing apples with oranges The analysis on the basin scale is focusing on a black and white example: a basin showing high correlation and therefore supporting the hypothesis of the authors and on the other hand a basin with low correlation contradicting the hypothesis (the mean correlation for all basins is 0.47 (se +/- 0.4) which is rather low). To resolve the low correlation in the black basin the authors split the data set according to a population by maximizing the correlation coefficient turning the black into a white one. Suggestion 6: Using the population as a basis for classifing rural and urban regions reminds me of using a dummy variable in regression from their it is only a slight jump to use population as variable in conjunction with preciptation. Using a ANOVA (or testing against a 0 slope of population or precipitation) would do the trick to see which one of the two is more important. But following suggestion 3 the influence of the population should vanish if and only if the hypothesis of a linear model in only influenced by precipitation is correct. The last subset of observation is the MAB (metropolian area of barcelona) suggesting that a finer temporal grain (30 min) of the precipitation is enhancing the predtiction of compensation payments. Then the precipitation is correlated with the precipitation in 24h which results in a low correlation. Now the whole other data analysis is based on the 24h precipitation but the 30 min seems
to be better suited. What are the implications for the 24h precipitation used for the other data sets? Suggestion 7: Presenting scatter plots are much better suited than maps in my humble opinion. The whole point of the study is the assumption of linearity between precipitation and compensation and simple plot could demonstrade this with elegant ease.

Response: First, the precipitation data went through a quality control process, only taking into account those stations with operations higher than 90% for the period of the study. In addition, different precipitation thresholds (for 24 h and 30 minutes in the case of the MAB) were tested in the model, and their results are shown (in the manuscript and in the supplementary material). As mentioned before, we considered the relative impacts of socio-economic factors on the damage in our models. That is, we consider three damage categories: total damage (D), damage per capita (DPC) and damage per unit of gross domestic product (DPW). For the MAB region we tested the model skill using two different time resolutions for precipitation data: 30 minutes and 24 hours. As shown in Figure 6 of the revised manuscript, the insurance data is more correlated with 30 minute precipitation. For this reason, we used this data in the logistic regression. Unfortunately this data is not available for all of Catalonia. Finally, following suggestions 4 and 7, we have added scatter plots to the manuscript (Figures 2 and 6) and the supplementary material (Figures 4, 5, 6, 7).

Referee's Comment: 3 Final Statement I hope the review was not unpolite and has in any way offented the auhtors which was not at all my purpose. I think the study needs a major overhaul regarding the data preprocessing as well as the techniques used to draw conclusions

Response: We want to show our sincere gratitude for all the comments and suggestions made by the reviewer. They have been very useful and constructive to make substantial improvements to the article.

References:
Bernet, D. B., Prasuhn, V. and Weingartner, R.: Surface water floods in Switzerland: What insurance claim records tell us about the damage in space and time, Nat. Hazards Earth Syst. Sci., 17(9), 1659–1682, doi:10.5194/nhess-17-1659-2017, 2017.

Falconer, R. H., Cobby, D., Smyth, P., Astle, G., Dent, J. and Golding, B.: Pluvial flooding: New approaches in flood warning, mapping and risk management, J. Flood Risk Manag., 2(3), 198–208, doi:10.1111/j.1753-318X.2009.01034.x, 2009.

Please also note the supplement to this comment: https://www.nat-hazards-earth-syst-sci-discuss.net/nhess-2017-278/nhess-2017-278-AC2-supplement.zip
**Fig. 1.** Map of Catalonia showing the aggregated basins, the Metropolitan Area of Barcelona (MAB), the main rivers and the pluviometric stations used.

---

## Author Comment (AC3) · 24 Nov 2017

Manuscript: nhess-2017-278 "The relationship between precipitation and insurance data for surface water floods in a Mediterranean region (Northeast Spain)".

Responses to reviewer #3:

Reviewer #3 (Article summary): The article analyses the correlation between extreme rainfall events and compensation costs triggered by flash floods, which are drawn from insurance records. The correlation coefficient is used to draw conclusions on the causal effect between precipitation and damage magnitude, using different scales

of aggregation as tests.

Response: We wish to thank the anonymous referee for his/her useful and constructive comments. Each specific point has been addressed in the manuscript as explained below.

Referee's Comment: The topic is of great importance and the use of empirical data is a plus, however the thinking behind the paper is a bit too much straightforward. Indeed precipitation is a major driver of flash flood damage; but it is not the only factor. The paper do not take in account other factors influencing damage (slopes, land cover and soil sealing, vegetation), and explains the effect (damage) by stating the cause (heavy rainfall); the conclusion uses correlation values to confirm the hypotesis.

Response: We would like to thank you for this very important comment. To address it, we applied a logistic regression model to gauge the probability of large damaging events given a certain precipitation amount, an approach that is frequently used for this kind of modelling study (Kim et al., 2012; Wobus et al., 2014). Since most of floods that affect this region are caused by in situ precipitation (surface water floods), our hypothesis is that precipitation is the main cause. We agree with the reviewer that other factors can influence the insurance compensations for floods. For this reason we now consider three damages categories: (i) total damages (D), (ii) damage per capita (DPC) and (iii) damage per unit of gross domestic product (DPW). In this way the relative impacts of socio-economic factors on damage can be estimated, while taking into account population and wealth (Zhou et al., 2017).

Referee's Comment: The statistical analysis needs to go deeper and to add more insights in relation to the distribution of damage along different typologies of exposure. The analysis uses 4 different aggregation scales based on admistrative units; most commonly in these kind of studies the scale would be smaller than the municipality. A projection of the data over built-up areas from land cover, building units or a regular grid cells would improve the analysis by linking the variables at a more detailed and
homogeneus unit compared to the administrative boundaries. I would suggest then to present only the results relative to the better performing aggregation method, as the comparison on administrative units do not produce added value for the conclusions.

Response: We have taken into account these useful comments on the methodology of our study. Our current model includes exposure in the damage data, as mentioned before. Following the suggestion of the reviewer, we have also aggregated the data on a basin level. In the supplementary material we also include the results based on the warning areas used by the Spanish State Meteorological Agency (AEMET). Our aim is not to estimate the precise amount of insurance payments made, but to estimate when a damaging event will occur given a certain precipitation amount. For this reason, we have applied a logistic model for different precipitation thresholds and types of damage in order to know when a damaging event occurs.

Referee's Comment: The difference between different kind of floods and to which kind exactly the compensatory records refer is not clearly stated in the paper. Overall, both the record data and the spatial data needs to be presented more precisely and clearly.

Response: In our model we have used all the flood events recorded in the INUNGAMA database. This database records the flood events that have affected Catalonia, most caused by in situ precipitation (surface water floods). For this reason, our hypothesis is that precipitation is the main cause of damaging floods. However, bearing in mind the possibility of having insurance data related with river floods, we used different precipitation thresholds in the model. In order to clarify this point, we added this sentence to the revised manuscript: "Most floods that have affected the region of study, Northeast Spain, are surface water floods. This type of floods can be regarded as coming under the most general definition of rainfall-related floods (Bernet et al., 2017), including pluvial floods but also flooding from sewer systems, small open channels, diverted watercourses or flooding from groundwater springs (Falconer et al., 2009). River floods that affect great distances are very rare in the region, and are only related to catastrophic and extended floods (for the analysed period only the October 2000
floods were of this type). Nevertheless, these are usually absorbed by reservoirs. It is therefore expected that flood insurance data will correlate strongly with precipitation and surface water floods."

Referee's Comment: Maps can be easily reduced in numbers and made more readable: figure 1 "a" and "b" can be combined by showing only the necessary information (river, basins, population, scores). Same goes for figure 3, it could be combined into 1 or 2 showing the information (dots) in different shapes/colors.

Response: We have followed the suggestion made by the reviewer and have changed the figures in the manuscript in order to make the paper more readable.

Referee's Comment: Finally, I agree with the insightful comments by reviewer 1 and 2 and suggest to majorly revise the paper by rethinking its objectives and methods.

Response: Taking into account all the comments and suggestions made by the reviewers, we have completely rewritten the manuscript, and we are confident that these major changes have improved the statistical significance of the analyses.

References:

Bernet, D. B., Prasuhn, V. and Weingartner, R.: Surface water floods in Switzerland: What insurance claim records tell us about the damage in space and time, Nat. Hazards Earth Syst. Sci., 17(9), 1659–1682, doi:10.5194/nhess-17-1659-2017, 2017.

Falconer, R. H., Cobby, D., Smyth, P., Astle, G., Dent, J. and Golding, B.: Pluvial flooding: New approaches in flood warning, mapping and risk management, J. Flood Risk Manag., 2(3), 198–208, doi:10.1111/j.1753-318X.2009.01034.x, 2009.

Kim, Y. O., Seo, S. B., & Jang, O. J. (2012). Flood risk assessment using regional regression analysis. Natural hazards, 63(2), 1203-1217.

Wobus, C., Lawson, M., Jones, R., Smith, J., & Martinich, J. (2014). Estimating monetary damages from flooding in the United States under a changing climate. Journal of Interactive comment
Flood Risk Management, 7(3), 217-229.

Zhou, Q., Leng, G. and Feng, L.: Predictability of state-level flood damage in the conterminous United States: the role of hazard, exposure and vulnerability, Sci. Rep., 7(1), 5354, doi:10.1038/s41598-017-05773-4, 2017.

Please also note the supplement to this comment: https://www.nat-hazards-earth-syst-sci-discuss.net/nhess-2017-278/nhess-2017-278-AC3-supplement.zip

**NHESSD**
**Fig. 1.** Map of Catalonia showing the aggregated basins, the Metropolitan Area of Barcelona (MAB), the main rivers and the pluviometric stations used.

---

## Referee Report (RR1)

**Article summary**

The article analyses the correlation between extreme rainfall events and compensation costs triggered by flash floods, which are drawn from insurance records. The correlation coefficient is used to draw conclusions on the causal effect between precipitation and damage to structures and infrastructures based on public insurance records.

**General comments**

The presentation of the case study, the statistical analysis and the overall quality of writing improved since the revision. The results confirm that precipitation is a key factor in explaining the damage caused by flash floods in regions these are the most common type of inundation. This self-evidence is stated more clearly from previous author comment, "Since most of floods that affect this region are caused by in situ precipitation (surface water floods), our hypothesis is that precipitation is the main cause." Also confirms that the damage is higher where the wealth is larger.

Since the regression analysis had rather poor results, I would have liked a better effort on the spatial disaggregation of data, in order to have a larger and more detailed sample than 29 basins. The inclusion of physical indicators such slope and vegetation could also help to characterize better the vulnerability in each basin. Then, a better testing of the hypotesis could be made on the relative importance of each factor as explanatory variables.

I suggest to read "Wagenaar et al. (2017) - Multi-variable flood damage modelling with limited data."

The paper is informative about the phenomena of flash floods in Catalonia, but I feel it does not add much value to the scientific knowledge on this field.

---

## Author Response (AR2)

**Manuscript: nhess-2017-278**
**OLD TITLE "The relationship between precipitation and insurance data for flood damages in a region of the Mediterranean (Northeast Spain)"**
**NEW TITLE "The relationship between precipitation and insurance data for floods in a Mediterranean region (Northeast Spain)"**

Dear editor,

In accordance with your instructions, we submit the revised version of our manuscript to NHESS, taking into account the editor's and reviewers' comments. They were very useful to improve this paper with a clearer definition of the type of floods we analysed, as well as a wider discussion, especially with regard to the scientific benefits of the study. They have also helped make the paper more readable.

The most evident changes applied to the manuscript are summarized below:
    i) We have re-written both the "Introduction", to state our scientific questions/hypotheses more clearly, and the "Conclusions" section, with a wider discussion of the potential applications of our results, the limitations of the methodology employed and possible future extension.
    ii) In the revised version of the manuscript, we removed the terms "flash floods" and we have re-organized the material in the manuscript accordingly.
    iii) Several sensitivity tests have been performed, confirming the main conclusion reported in the previous version of the manuscript.

We are confident that these changes have improved the statistical significance of the analyses as well as the manuscript's clarity. These changes have not altered our conclusions and key findings. Instead, they have strengthened the main message of the previous version.

We wish to thank the editor and reviewers for their time and their suggestions for improving the manuscript.

We enclose a letter with detailed responses to the reviewers' comments and showing where changes have been made to the manuscript.

We hope this revised manuscript now meets the NHESS criteria for publication.

Sincerely,

Maria Cortès on behalf of the co-authors

**Manuscript: nhess-2017-278**
**OLD TITLE "The relationship between precipitation and insurance data for flood damages in a region of the Mediterranean (Northeast Spain)"**
**NEW TITLE "The relationship between precipitation and insurance data for floods in a Mediterranean region (Northeast Spain)"**

**Responses to reviewer #1:**

**Reviewer #1 (Highlight):** *The manuscript improved remarkably in the updated version. However, some weak points remain in the manuscripts.*
**Response:** We wish to thank the anonymous referee for his/her useful and constructive comments. Each specific point has been addressed in the manuscript as explained in the following document.

**Referee's Comment***: While the hydrologic process that is investigated now has been defined more strictly (surface water floods), the differentiation between "flash floods" remain unclear. Are these two terms used synonymously?*
**Response:** We agree with the reviewer that the terms "surface water floods" and "flash floods" may confuse the reader. In the revised version of the manuscript, we have removed the terms "flash floods" and we have re-organised the material in the manuscript accordingly.

**Referee's Comment***: As in the first version the focus laid on floods in general and now the focus is on surface water floods, I am wondering how the authors created the subset of claims affected exclusively by this type of floods from the whole set of flood losses.*
**Response:** Unfortunately, the database does not provide any information on the type of flood. However, it is worth noting that most of the floods that affect the region of study are surface water floods, while river floods are very rare in the region, and only occur when related to catastrophic and extended floods (see e.g. Llasat et al., 2014, 2016a). To give an example, the October 2000 floods were the only time this type of flood occurred during the analysed period.

**Referee's Comment***: The main goal and the research question remain unclear. For me, it is not clear whether the main aim is to search for a threshold, or to forecast losses, etc. I highly recommend to sharpen the research question.*
**Response:** We wish to thank the reviewer, whose comment provides us with further ways in which we can explain the added value of our study. We have thus re-written both the "Introduction" section, to state our scientific questions/hypotheses more clearly, and the "Conclusions" section, with wider discussion of the potential applications of our results, the limitations of the methodology employed, and possible future extension.

**Referee's Comment***: It is not clearly described, how precipitation in Fig. 2 is calculated. Is the value for precipitation an average over Catalonia or it is the maximum value of all meteorological stations in Catalonia? The method for combining pairs of precipitation values and claims has to be described in more detail, especially in regards to the spatial reference unit at which aggregation of both variables was made.*
**Response:** We have changed the caption in Figure 2 to:

"Scatter plot showing maximum precipitation in 24 h (mm) and (a) total damages (D); (b) damage per capita (DPC); and (c) damage per unit of wealth (DPW), for flood events recorded in Catalonia between 1996 and 2015 (log-transformed values; damage are given in euros). Each point represents the insurance compensation series (D, DPC or DPW) and the maximum 24 h precipitation for each basin. The dashed line indicates the fit based on a linear regression model."

**Referee's Comment**: *In Fig. 2 it is not clear if the units for precipitation are in [mm] or in [mm/h] (valid also for Fig. 4, 6 and 7).*
**Response:** We have now specified, in the caption of Figs. 2, 4, 6 and 7, that precipitation is given in mm.

**Referee's Comment**: *The authors state that exact data on the value and location of assets exposed are not available (p. 30, ln. 2). However, in chapter 2.2 they state that the claims are aggregated on postcode level. Since the number of rainfall gauging stations is relatively dense in the lowland areas, the minimal spatial reference unit for analysing pairs of records should be the postcode area. Precipitation can be interpolated between stations to cover postcode areas without rainfall gauging stations. However, the mountain regions might be better excluded from the analysis. This corresponds with a statement that coastal areas are the most affected by flood events.*
**Response:** We have analysed the precipitation-compensation link for each basin in Catalonia, since the number of floods is too sparse to support our statistical assessment on a postcode scale.
We have also assessed the robustness of our results, carrying out a number of sensitivity checks: considering (i) a precipitation threshold of 40 mm instead of 60 mm, (ii) the AEMET warning areas, (iii) only coastal regions and (iv) the basin averaged precipitation instead of the maximum values. Overall, choosing different methods led to a similar conclusion. These results are shown in the supplementary material.

**Referee's Comment**: *eIn chapter 2.2. (p. 2, ln 15.) the authors write that they compare the maximum of precipitation measured in a basin with losses aggregated at basin level. As some basins have a relatively heterogeneous topography and a remarkably aerial size, and as the main focus is on surface water floods, this assumption for correlation might be misleading. An intensive precipitation event captured by one of the rainfall stations can be restricted to a few kilometres and may not be representative for the whole basin. Thus, instead of the maximum precipitation an average over the whole basin might be more relevant when correlating precipitation with losses aggregated over the whole basin. The focus on the MAB underlines this. Another option is to focus on smaller spatial reference units as the post code areas, as proposed above. Furthermore, the paper could also be more sophisticated if it focuses on MAB only.*
**Response:** We have taken into account the suggestions made by the reviewer and added the results using the average precipitation of the flood case in the basin. As the new results are similar to the previous ones, the relationship with average precipitation is shown in the supplementary material. See also the response to the previous referee's comment.

**Referee's Comment**: *Regarding the calculation of the hit rate and false alarm rate, it would be very useful to define more clearly what has been count as a hit and what has been count as a false alarm. Furthermore, it has to be defined how the validation metrics are calculated*

*and on which sample (how was the validation sample extracted from the whole data set and how many records it contains).*

**Response:** We added the "Verification method" section to clarify the calculation of the hit rate and false alarm rate.

**OLD TITLE "The relationship between precipitation and insurance data for flood damages in a region of the Mediterranean (Northeast Spain)"**
**NEW TITLE "The relationship between precipitation and insurance data for floods in a Mediterranean region (Northeast Spain)"**

**Responses to reviewer #2:**

**Reviewer #2 (Highlight):** *I liked how you (the authors) changed the flow of teh article and you really improved it from the last version. The data now seems more precise and clearer also the new maps enhance the understanding of the text.*
**Response:** We wish to thank the anonymous referee for his/her useful and constructive comments. Each specific point has been addressed in the manuscript as explained in the following document.

**Referee's Comment**: *Still - and I am really sorry to say that - the article misses some meat. The main outcome is still yes Rainfall has an influence or more precise the rainfall has an influence on the damage in the sense that it explains about 10 to 15% of the variability of the damage data. Still not a precise result but better than simple guessing aye?*
**Response:** As we better articulate in the revised version of the manuscript, our main goal is to develop a parsimonious model to predict flood damages. On the one hand, this analysis would provide information on the links between floods and their drivers in Mediterranean regions. On the other hand, the models produced could allow us to predict when damaging events (with its meaning depending on the user) will occur as a result of a certain precipitation threshold. In addition, these models could be applied to other areas, and could be used to predict flood damage in future climate change scenarios.
Regarding the 10 to 15% variance explained, we actually show the scatter plots in Figure 2 to illustrate that, although precipitation has a statistically significant influence on damage, the explanatory power of a linear regression model is rather low. That is, modelling insurance compensations is a complex issue, due to the limitations in observational data and the concurrence of a variety of relevant factors. Nevertheless, the significant correlation between insurance compensations and precipitation suggests that rainfall data can be used to extract information on damage in Catalonia. To do this, we applied a logistic regression model to gauge the probability of large damaging events occurring given a certain precipitation amount.

**Referee's Comment**: *The logistic model is a nice idea but what to we lern from it? A somehow apritrary chossen threshhold (whats a amount of money that really hurts?) has a weak probability dependency on rainfall. Okay but whats next? The article you have cited in the context was using it to predict how climate change may alter the amount of damage but in this case…*
*Okay thats a lot of critique but to I have some suggestions? Maybe one could elaborate on the uncertainty in the quantile? How exact is its value? How does the model change with new samples? Are their more predictors that could explain the damage? What do we do with that results?*
*I know this is a little of a downer because I can see and literally feel the effort that went into that revision but still as I mentioned above their is something missing.*

**Response:** In the revised version of the manuscript we have re-written both the "Introduction" section, to state our scientific questions/hypotheses more clearly, and the "Conclusions" section, with wider discussion of the potential applications of our results, the limitations of the methodology employed and possible future extension. Regarding the percentiles, we consider different thresholds to identify significant damage, since the meaning of "significant damage" depends on the user. Additionally, we tested two different thresholds to define heavy precipitation, as well as showing three different categories of damages, and a number of sensitivity checks: (i) we repeated the analysis the AEMET warning areas, (ii) considering only coastal regions and (iii) the basin averaged precipitation instead of the maximum values. Overall, choosing different method's choices lead to a similar conclusion. These results are shown in the supplementary material.

**Manuscript: nhess-2017-278**
**OLD TITLE "The relationship between precipitation and insurance data for flood damages in a region of the Mediterranean (Northeast Spain)"**
**NEW TITLE "The relationship between precipitation and insurance data for floods in a Mediterranean region (Northeast Spain)"**

**Responses to reviewer #3:**

**Reviewer #3 (Highlight):** *General comments*
*The presentation of the case study, the statistical analysis and the overall quality of writing improved since the revision. The results confirm that precipitation is a key factor in explaining the damage caused by flash floods in regions these are the most common type of inundation. This self-evidence is stated more clearly from previous author comment, "Since most of floods that affect this region are caused by in situ precipitation (surface water floods), our hypothesis is that precipitation is the main cause." Also confirms that the damage is higher where the wealth is larger.*
**Response:** We wish to thank the anonymous referee for his/her useful and constructive comments. Each specific point has been addressed in the manuscript as explained in the following document.

**Referee's Comment 1**: *However, it does not inclued physical characteristics (such as slope, soil characteristics, vegetation) in the analysis. The regression model shown in figure 2 has rather poor explanatory value. Regression is used to prove causation.*
**Response:** As we better articulate in the revised version of the manuscript, our main goal is to develop a parsimonious model to predict flood damages, using precipitation as the only predictor variable. However we have re-written the "Conclusions" section, with wider discussion of the potential applications of our results, the limitations of the methodology employed and possible future extension:

> "In addition, more complex analyses including more sophisticated empirical methods, and other factors such as soil physical characteristics (e.g. slope, soil characteristics, vegetation), could provide additional understanding on flood drivers and impacts"

We show the scatter plots in Figure 2 to illustrate that, although, precipitation has a statistically significant influence on the damage, the explanatory power of a linear regression model is rather low. That is, modelling insurance compensations is a complex issue, due to the limitations in observational data and the concurrence of a variety of relevant factors. Nevertheless, the significant correlation between insurance compensations and precipitation suggests that rainfall data can be used to extract information on damage in Catalonia. To do this, we applied a logistic regression model to gauge the probability of large damaging events occurring given a certain precipitation amount.

**Referee's Comment 2**: *Another problem is the broad scale of spatial aggregation (29 very heterogeneous basins) used in the analysis that do not allow to get a large sample, providing more detailed insights which would be expected since the case study area is sub-national (1 region).*
**Response:** Actually, the broad scale of spatial aggregation used allowed us to have quite a large sample size, with 239 flood cases (see Table 2). As explained in Section 2.2, we use the

expression "flood case" for each pair of values corresponding to a basin affected by a flood event.

**Referee's Comment 3**: *Article summary*
*The article analyses the correlation between extreme rainfall events and compensation costs triggered by flash floods, which are drawn from insurance records. The correlation coefficient is used to draw conclusions on the causal effect between precipitation and damage to structures and infrastructures based on public insurance records.*
**Response:** Actually, we do not use correlation to draw conclusions on the link between precipitation and damages (see also the response to referee comment 1).

**Referee's Comment 4**: *Since the regression analysis had rather poor results, I would have liked a better effort on the spatial disaggregation of data, in order to have a larger and more detailed sample than 29 basins. The inclusion of physical indicators such slope and vegetation could also help to characterize better the vulnerability in each basin. Then, a better testing of the hypotesis could be made on the relative importance of each factor as explanatory variables.*
*I suggest to read "Wagenaar et al. (2017) - Multi-variable flood damage modelling with limited data."*
*The paper is informative about the phenomena of flash floods in Catalonia, but I feel it does not add much value to the scientific knowledge on this field.*
**Response:** First, as explained in the response to referee comment 2, linear regression analysis is shown to illustrate that modelling insurance compensations is a complex issue, and the logistic regression model makes it possible to extract information from precipitation records on damage in Catalonia.

As explained in the response to referee comment 2, the spatial aggregation used allowed us to have quite a large sample size, with 239 flood cases (see Table 2). In addition, we also repeated the analysis considering a different spatial aggregation, the AEMET warning areas, and obtained very similar results.

We have also better articulated the added value of our study. Specifically, to the best of our knowledge, this is one only a few studies on the possible links between precipitation and economic flood damage in a Mediterranean region. We have thus re-written both the "Introduction" section, to state our scientific questions/hypotheses more clearly, and the "Conclusions" section, with wider discussion of the potential applications of our results, the limitations of the methodology employed and possible future extension.

[revised manuscript text omitted]

---

## Author Response (AR3)

**Manuscript: nhess-2017-278**
**"The relationship between precipitation and insurance data for floods in a Mediterranean region (Northeast Spain)"**

Dear editor,

In accordance with your instructions, we submit the revised version of the manuscript nhess-2017-278 to be considered for publication in NHESS, taking into account the editor's comments.

Following your recommendations, we have changed the section 4 of our manuscript (Conclusions) and we have integrated a discussion in it (Discussion and conclusions), comparing our results with other papers related with insurance data and floods.

We thank the editor and reviewers for their time and their suggestions for improving the manuscript.

We hope this revised manuscript now meets the NHESS criteria for publication.

Sincerely,

Maria Cortès on behalf of the co-authors

[revised manuscript text omitted]
 and this result confirms previous studies such the one of Torgersen et al. (2015), that have found a significant relation between insurance data and short time duration rainfall when studying urban floods in Norway. In addition, Spekkers et al. (2013) have showed that high claim numbers associated to private property and content damage, were significantly related to maximum rainfall intensity, based on a logistic regression, with rainfall intensity for 10-min to 4-h time windows.

Overall, our results confirm the hypothesis that precipitation is a key factor in explaining the damage caused by flood events in regions in which water surface floods are the main type of floods, as is the case in the Mediterranean region of study. Also our findings align with the results of previous studies (Spekkers et al., 2013; Zhou et al., 2013; Wobus et al., 2014; Torgersen et al., 2015) and further indicate that insurance databases are a promising source for flood damage assessment at local (Garrote et al., 2016; Le Bihan et al., 2017; Ziscgh et al., 2018; Zhou et al., 2013) and at regional scale (Barredo et al., 2012; Kim et al., 2012; Wobus et al., 2014; Zhou et al., 2017).

To summarize, we have developed a new model that allows us to predict the probability that a flood event causing large damages (where the meaning of "large" depends on the user) will occur, based on precipitation, and taking into account the exposure and vulnerability of the region in the model. That is,

5 the parsimonious empirical models linking flood damages to heavy precipitation developed in this study are a step towards providing a substantial contribution to developing a warning forecast system with flood management strategies. For instance, from the relationship shown between precipitation and insurance compensations it is possible to predict when damaging events

10 will occur as a result of a certain precipitation threshold.

These results were obtained by following a simple and transparent statistical methodology that can also be applied to other areas. These links could also provide a basis to predict flood damage in future climate change scenarios as done for instance

15 by Wobus et al. (2014) that estimated monetary damages from flooding in the United States under a "business as usual" climate change scenario. As a word of caution it is worth noting that the complex relationships between climate variability, human activities and flood damages may limit the applicability of these findings to conditions that are very different from current ones. In addition, more complex analyses including more sophisticated empirical methods, and other factors such as soil physical characteristics (e.g. slope, soil characteristics, vegetation), could provide additional understanding on flood drivers

20 and impacts. For instance, in Garrote et al. (2016) different simulation scenarios were defined considering the modifications to the terrain due to construction of fluvial defence structures in the area.

Despite these limitations, this work has provided the first assessment of the link between precipitation and flood damages in a Mediterranean region, and our results suggest that by exploiting the relationship between precipitation and flood damages, the

25 model could provide a satisfactory prediction of monetary compensation.

**Competing interests**

The authors declare that they have no conflicts of interest.

**Acknowledgments**

This work has been supported by the Spanish Project HOPE (CGL2014-52571-R) of the Ministry of Economy, Industry and Competitiveness, the Metropolitan Area of Barcelona Project (no. 308321) (Flood evolution in the Metropolitan Area of Barcelona from a holistic perspective: past, present and future) and the Water Research Institute (IdRA) of the University of Barcelona. It was conducted under the framework of the HyMeX Programme (HYdrological cycle in the Mediterranean EXperiment) and the Panta Rhei WG Changes in Flood Risk. We would like to thank AEMET and SMC for the meteorological and hydrological information provided for this study. Thanks also to BCASA for the detailed information about the system used to prevent and manage floods. Marco Turco was supported by the Spanish Juan de la Cierva Programme (IJCI-2015-26953). We would also like to acknowledge Hannah Bestow for correcting the English language of this paper.

**References**

[revised manuscript text omitted]